# RAG1 and RAG2 non-core regions are implicated in leukemogenesis and off-target V(D)J recombination in BCR-ABL1-driven B-cell lineage lymphoblastic leukemia

Xiaozhuo Yu[1†], Wen Zhou[1†], Xiaodong Chen[1], Shunyu He[1], Mengting Qin[1], Meng Yuan[1], Yang Wang[1], Woodvine Otieno Odhiambo[1], Yinsha Miao[2]*, Yanhong Ji[1,2]*

[1]Department of Pathogenic Biology and Immunology, School of Basic Medical Sciences, Xi'an Jiaotong University Health Science Center, Xi'an, China; [2]Department of Clinical Laboratory, Xi'an No. 3 Hospital, the Affiliated Hospital of Northwest University, Xian, China

*For correspondence:
miaoyinsha@med.nwu.edu.cn (YM);
jiyanhong@xjtu.edu.cn (YJ)

[†]These authors contributed equally to this work

**Abstract** The evolutionary conservation of non-core RAG regions suggests significant roles that might involve quantitative or qualitative alterations in RAG activity. Off-target V(D)J recombination contributes to lymphomagenesis and is exacerbated by RAG2' C-terminus absence in $Tp53^{-/-}$ mice thymic lymphomas. However, the genomic stability effects of non-core regions from both $Rag1^{c/c}$ and $Rag2^{c/c}$ in $BCR\text{-}ABL1^+$ B-lymphoblastic leukemia ($BCR\text{-}ABL1^+$ B-ALL), the characteristics, and mechanisms of non-core regions in suppressing off-target V(D)J recombination remain unclear. Here, we established three mouse models of $BCR\text{-}ABL1^+$ B-ALL in mice expressing full-length RAG ($Rag^{f/f}$), core RAG1 ($Rag1^{c/c}$), and core RAG2 ($Rag2^{c/c}$). The $Rag^{c/c}$ ($Rag1^{c/c}$ and $Rag2^{c/c}$) leukemia cells exhibited greater malignant tumor characteristics compared to $Rag^{f/f}$ cells. Additionally, $Rag^{c/c}$ cells showed higher frequency of off-target V(D)J recombination and oncogenic mutations than $Rag^{f/f}$. We also revealed decreased RAG cleavage accuracy in $Rag^{c/c}$ cells and a smaller recombinant size in $Rag1^{c/c}$ cells, which could potentially exacerbate off-target V(D)J recombination in $Rag^{c/c}$ cells. In conclusion, these findings indicate that the non-core RAG regions, particularly the non-core region of RAG1, play a significant role in preserving V(D)J recombination precision and genomic stability in $BCR\text{-}ABL1^+$ B-ALL.

## eLife assessment

Using a set of animal models, this **valuable** paper shows tumor suppressive function of the non-core regions of RAG1/2 recombinases. The conclusions are supported by **solid** evidence.

## Introduction

V(D)J recombination serves as the central process for early lymphocyte development and generates diversity in antigen receptors. This process involves the double-strand DNA breaks of gene segments by the V(D)J recombinase, including Rag1 and Rag2 (hereinafter referred to as RAG). RAG recognizes conserved recombination signal sequences (RSSs) positioned adjacent to V, D, and J gene segments. A bona fide RSS contains a conserved palindromic heptamer (consensus 5'-CACAGTG) and A-rich

nonamer (consensus 5′-ACAAAAACC) separated by a degenerate spacer of either 12 or 23 base pairs (*Hirokawa et al., 2020*; *Schatz and Ji, 2011a*). The process of efficient recombination is contingent upon the presence of RSSs with differing spacer lengths, as dictated by the '12/23 rule' (*Banerjee and Schatz, 2014*; *Eastman et al., 1996*; *Shi et al., 2020*). Following cleavage, the DNA ends are joined via non-homologous end-joining (NHEJ), resulting in the precise alignment of the two coding ends and the signal ends (*Rooney et al., 2004*). V(D)J recombination promotes B-cell development, but aberrant V(D)J recombination can lead to precursor B-cell malignancies through RAG-mediated off-target effects (*Mendes et al., 2014*; *Onozawa and Aplan, 2012*; *Thomson et al., 2020*).

The regulation of RAG expression and activity is multifactorial, serving to ensure V(D)J recombination and B-cell development (*Gan et al., 2021*; *Kumari et al., 2021*). The RAGs consist of core and non-core region. Although non-core regions of Rag1/2 are not strictly required for V(D)J recombination, the evolutionarily conserved non-core RAG regions indicate their potential significance in vivo that may involve quantitative or qualitative modifications in RAG activity and expression (*Braams et al., 2023*; *Curry and Schlissel, 2008*; *Liu et al., 2022*; *Liu et al., 2022*; *Sekiguchi et al., 2001*). Specifically, the non-core Rag2 region (amino acids 384–527 of 527 residues) contains a plant home-odomain (PHD) that can recognize histone H3K4 trimethylation, as well as a T490 locus that mediates a cell cycle-regulated protein degradation signal in proliferated pre-B-cell stage (*Liu et al., 2007*; *Matthews et al., 2007*). Failure to degrade Rag2 during the S stage poses a threat to the genome (*Zhang et al., 2011*). Moreover, the off-target V(D)J recombination frequency is significantly higher when Rag2 is C-terminally truncated, thereby establishing a mechanistic connection between the PHD domain, H3K4me3-modified chromatin, and the suppression of off-target V(D)J recombination (*Lu et al., 2015*; *Mijušković et al., 2015*). The Rag1' non-core region (amino acids 1–383 of 1040 residues) has been identified as a Rag1 regulator. While the core Rag1 maintains its catalytic activity, its in vivo recombination efficiency and fidelity are reduced in comparison to the full-length Rag1 (*Silver et al., 1993*). In addition, the Rag1 binding to the genome is more indiscriminate (*Beilinson et al., 2021*; *Sadofsky et al., 1993*). The N-terminal domain (NTD), which is evolutionarily conserved, is predicted to contain multiple zinc-binding motifs, including a Really Interesting New Gene (RING) domain (aa 287–351) that can ubiquitylate various targets, including Rag1 itself (*Deng et al., 2015*). Furthermore, NTD contains a specific region (amino acids 1–215) that facilitates interaction with DCAF1, leading to the degradation of Rag1 in a CRL4-dependent manner (*Schabla et al., 2018*). Additionally, the NTD plays a role in chromatin binding and the genomic targeting of the RAG complex (*Schatz and Swanson, 2011b*). Despite increased evidence emphasizing the significance of non-core RAG regions, particularly Rag1's non-core region, the function of non-core RAG regions in off-target V(D)J recombination and the underlying mechanistic basis have not been fully clarified.

Typically, genomic DNA is safeguarded against inappropriate RAG cleavage by the inaccessibility of cryptic RSSs (cRSSs), which are estimated to occur once per 600 base pairs (*Teng et al., 2015*). However, recent research has demonstrated that epigenetic reprogramming in cancer can result in heritable alterations in gene expression, including the accessibility of cRSSs (*Becker et al., 2020*; *Fatma et al., 2022*; *Goel et al., 2022*; *Khoshchehreh et al., 2019*). We selected the *BCR-ABL1*⁺ B-ALL model, which is characterized by ongoing V(D)J recombinase activity and *BCR-ABL1* gene rearrangement in pre-B leukemic cells (*Schjerven et al., 2017*; *Wong and Witte, 2004*). The genome structural variations (SVs) analysis was conducted on leukemic cells from *Rag*^f/f^, *Rag1*^c/c^, and *Rag2*^c/c^, *BCR-ABL1*⁺ B-ALL mice to examine the involvement of non-core RAG regions in off-target V(D)J recombination events. The non-core domain deletion in both Rag1 and RAG2 led to accelerated leukemia onset and progression, as well as an increased off-target V(D)J recombination. Our analysis showed a reduction in RAG cleavage accuracy in *RAG*^c/c^ cells and a decrease in recombinant size in *RAG1*^c/c^ cells, which may be responsible for the increased off-target V(D)J recombination in *RAG*^c/c^ leukemia cells. In conclusion, our results highlight the potential importance of the non-core RAG region, particularly RAG1's non-core region, in maintaining accuracy of V(D)J recombination and genomic stability in *BCR-ABL1*⁺ B-ALL.

## Results

### *Rag^c/c* give more aggressive leukemia in a mouse model of *BCR-ABL1^+* B-ALL

In order to assess the impact of RAG activity on the clonal evolution of *BCR-ABL1^+* B-ALL through a genetic experiment, we utilized bone marrow transplantation to compare disease progression in *Rag^f/f*, *Rag1^c/c*, and *Rag2^c/c* *BCR-ABL1^+* B-ALL (*Schjerven et al., 2017*; *Wong and Witte, 2004*). Bone marrow cells transduced with a BCR-ABL1/green fluorescence protein (GFP) retrovirus were administered into syngeneic lethally irradiated mice, and CD19^+ B-cell leukemia developed within 30–80 days (*Figure 1A*, *Figure 1—figure supplement 1*). Western blotting results confirmed equivalent transduction efficiencies of the retroviral *BCR-ABL1* in all three cohorts (*Figure 1—figure supplement 2A*). To investigate potential variances in leukemia outcome across different genomic backgrounds, we employed Mantel–Cox analysis to evaluate survival rates in *Rag^f/f*, *Rag1^c/c*, or *Rag2^c/c* mice transplanted with *BCR-ABL1*-transformed bone marrow cells. Our results show that, during the primary transplant phase, *BCR-ABL1^+* B-ALL mice expressing Rag1^c/c or Rag2^c/c demonstrated lower survival rates compared to their counterparts with Rag^f/f (median 74.5 days vs 39 or 57 days, $p < 0.0425$, *Figure 1A*). This survival rates discrepancy was also observed during the secondary transplant phase, wherein leukemic cells were extracted from the spleens of primary recipients and subsequently purified via GFP^+ cell sorting. A total of $10^5$, $10^4$, and $10^3$ GFP^+ leukemic cells that originated from *Rag^f/f*, *Rag1^c/c*, or *Rag2^c/c* leukemic mice were transplanted into corresponding non-irradiated immunocompetent syngenetic recipients (survival days *Rag^f/f*, 11–26 days, *Rag1^c/c*, 10–16 days, *Rag2^c/c*, 11–21 days, *Figure 1—figure supplement 2B*). Additionally, the *Rag^c/c* mice exhibited significantly higher leukemia burdens in the peripheral blood, bone marrow, and spleen compared to the *Rag^f/f* mice (*Figure 1B, D*). To elucidate the cellular mechanisms driving the accelerated proliferation observed in *Rag^c/c* *BCR-ABL1^+* B-ALL, flow cytometry analyses were conducted to evaluate cell cycle dynamics and apoptotic activity. Results revealed a higher fraction of *Rag^c/c* *BCR-ABL1^+* B-ALL cells residing in the S/G2-M phase of the cell cycle compared to their *Rag^f/f* counterparts (*Figure 1E*). Additionally, the enhanced proliferation in *Rag^c/c* leukemic cells was attributed to a reduction in apoptosis rates (*Figure 1—figure supplement 2C*). RNA-seq analysis demonstrated the changes of cell differentiation and proliferation/apoptotic pathways (*Figure 1—figure supplement 3*) These findings indicate that the absence of non-core RAG regions accelerates malignant transformation and leukemic proliferation, leading to a more aggressive disease phenotype in the *Rag^c/c* *BCR-ABL1^+* B-ALL mouse model.

### The loss of non-core RAG regions corresponds to a less mature cell surface phenotype but does not impede IgH VDJ recombination

To delineate the developmental stages of B cells from which the leukemic cells originated, we performed flow cytometry on single cells stained with B-cell-specific surface markers. Analysis revealed that 91–98% of GFP^+ cells in *Rag^c/c* mice were CD19^+BP-1^+B220^+CD43^+, indicating that most leukemic cells were at the large pre-B-cell stage (*Figure 2A*; *Hardy and Hayakawa, 2001*). Conversely, in *Rag^f/f* leukemic mice, the distribution was 65% large pre-B cells (GFP^+CD19^+BP-1^+B220^+CD43^+) and 35% small pre-B cells (GFP^+CD19^+BP-1^+B220^+CD43^−) (*Figure 2A*). Moreover, approximately 5% of leukemic cells in *Rag^f/f* mice expressed μHC, in contrast to minimal expression in *Rag^c/c* leukemic cells. This suggests that *Rag^f/f* leukemic cells may differentiate further, associated with an immune phenotype (*Figure 2B*). IgH rearrangement initiates with $D_H$–$J_H$ joining in pro-B cells, followed by $V_H$–$DJ_H$ joining in pro-B cells, and ultimately, $V_L$–$J_L$ rearrangements occur at the *IgL* loci in small pre-B cells (*Schatz and Ji, 2011a*). Genomic polymerase chain reaction analysis of DNA from GFP^+CD19^+ cells was utilized to assess $V_H DJ_H$ rearrangement. The results showed a pronounced oligoclonality in *Rag^c/c* leukemic cells, with tumors consistently demonstrating rearrangements involving a restricted set of $V_H$ family members. In contrast, *Rag^f/f* leukemias displayed significant polyclonality, evidenced by the widespread rearrangement of various $V_H$ family members to all potential $J_H$1–3 segments, indicative of a broader clonal diversity (*Figure 2—figure supplement 1*). This observation aligns with the more aggressive leukemia phenotype seen in *Rag^c/c* *BCR-ABL1^+* B-ALL mice. Such oligoclonality in *Rag^c/c* leukemic cells suggests a selection process driven by BCR-ABL1-induced leukemia, favoring the emergence of a limited number of dominant leukemic clones. The absence of non-core RAG regions appears to restrict the diversity of leukemic clones, leading to the formation of oligoclonal tumors.

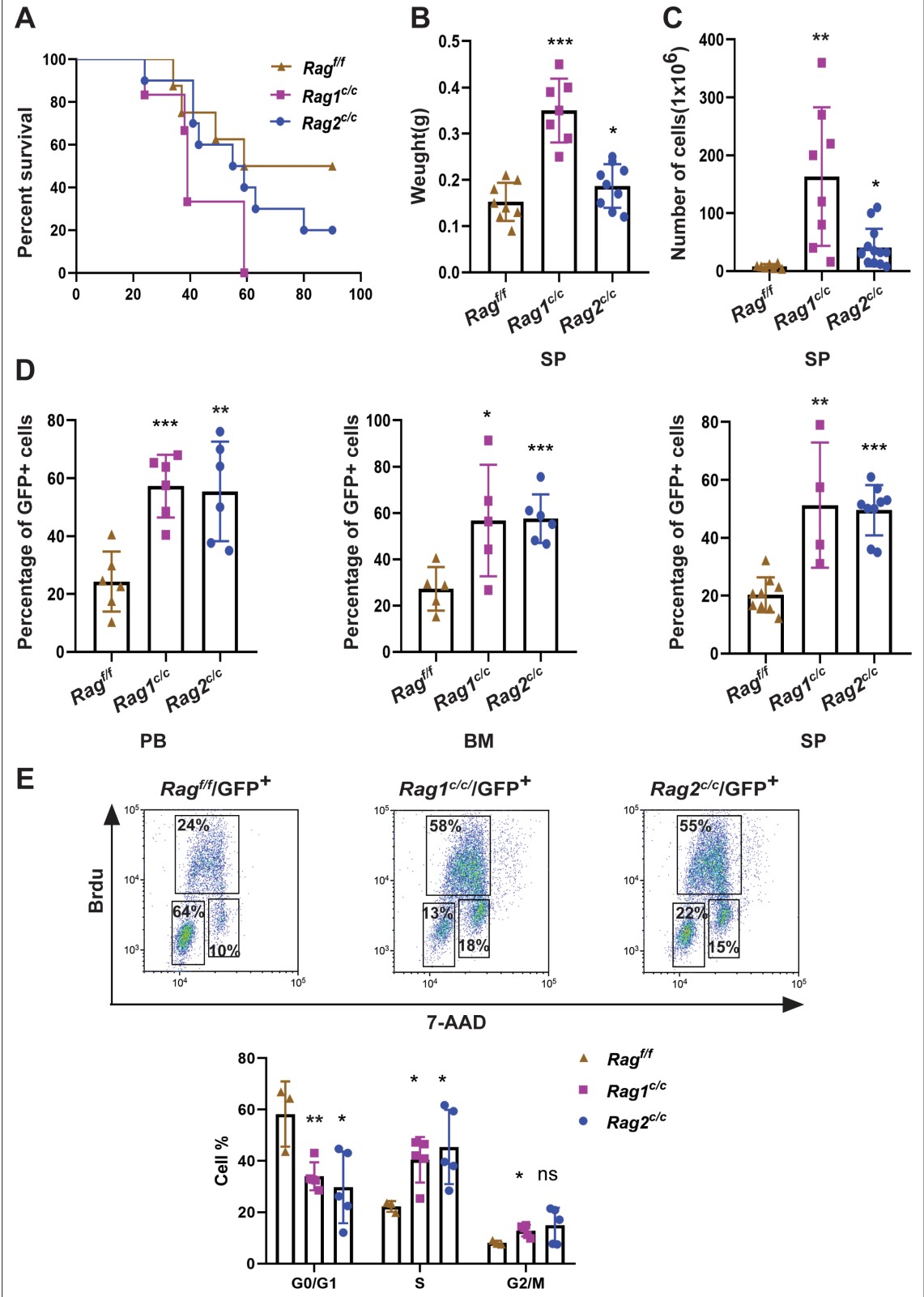

**Figure 1.** RAG$^{c/c}$s give more aggressive leukemia in mice model of *BCR-ABL1*$^+$ B-ALL. (**A**) Kaplan–Meier survival curve for *Rag*$^{f/f}$ (n = 8), *Rag1*$^{c/c}$ (n = 6), and *Rag2*$^{c/c}$ (n = 10) recipient mice. The survival was calculated by Mantel–Cox test (*Rag*$^{f/f}$ vs *Rag1*$^{c/c}$, p < 0.0381, *Rag*$^{f/f}$ vs *Rag2*$^{c/c}$, p < 0.0412). (**B**) The spleen weights of *Rag*$^{f/f}$, *Rag1*$^{c/c}$, and *Rag2*$^{c/c}$ leukemic mice (*Rag*$^{f/f}$, n = 8, *Rag1*$^{c/c}$, n = 7, *Rag2*$^{c/c}$, n = 9; *Rag*$^{f/f}$ vs *Rag1*$^{c/c}$ p < 0.0001, *Rag*$^{f/f}$ vs *Rag2*$^{c/c}$, p = 0.1352). (**C**) The numbers of spleen cell in *Rag*$^{f/f}$, *Rag1*$^{c/c}$, and *Rag2*$^{c/c}$ leukemic mice (*Rag*$^{f/f}$, n = 7, *Rag1*$^{c/c}$, n = 8, *Rag2*$^{c/c}$, n = 13; *Rag*$^{f/f}$ vs *Rag1*$^{c/c}$, p =

*Figure 1 continued on next page*

*Figure 1 continued*

0.0047, *Rag^f/f* vs *Rag2^c/c*, p = 0.0180). (**D**) The percentage of GFP⁺ cells in peripheral blood (PB) (*Rag^f/f*, n = 6, *Rag1^c/c*, n = 6, *Rag2^c/c*, n = 6; *Rag^f/f* vs *Rag1^c/c*, p = 0.0003, *Rag^f/f* vs *Rag2^c/c*, p = 0.0035), bone marrow (BM, *Rag^f/f*, n = 5, *Rag1^c/c*, n = 5, *Rag2^c/c*, n = 6; *Rag^f/f* vs *Rag1^c/c*, p = 0.0341, *Rag^f/f* vs *Rag2^c/c*, p = 0.0008), and spleen (SP, *Rag^f/f*, n = 9, *Rag1^c/c*, n = 4, cRAG2, n = 9; *Rag^f/f* vs *Rag1^c/c*, p = 0.0016, *Rag^f/f* vs *Rag2^c/c*, p < 0.0001) of *Rag^f/f*, *Rag1^c/c*, and *Rag2^c/c* leukemic mice. (**E**) Representative flow cytometry plots of cell cycle arrest of leukemic cells in *Rag^f/f*, *Rag1^c/c*, and *Rag2^c/c* mice. In the graph, the percentages of each phase of the cell cycle are summarized below (*Rag^f/f*, n = 3, *Rag1^c/c*, n = 5, *Rag2^c/c*, n = 5; G0/G1, *Rag^f/f* vs *Rag1^c/c*, p = 0.0082, *Rag^f/f* vs *Rag2^c/c*, p = 0.0279; S, *Rag^f/f* vs *Rag1^c/c*, p = 0.0146, *Rag^f/f* vs *Rag2^c/c*, p = 0.0370; G2/M, *Rag^f/f* vs *Rag1^c/c*, p = 0.0134, *Rag^f/f* vs *Rag2^c/c*, p = 0.1507). In figures B, C, D, and J, error bars represent the mean ± standard deviation (s.d.), p values were calculated by Student's *t* test and *p < 0.05, **p < 0.01, ***p < 0.001.

The online version of this article includes the following source data and figure supplement(s) for figure 1:

**Figure supplement 1.** Construction of *Rag^f/f*, *Rag1^c/c*, and *Rag2^c/c*, BCR-ABL1⁺ B-ALL mice models using bone marrow transplantation.

**Figure supplement 2.** Biological behavior of leukemia in *Rag^f/f*, *Rag1^c/c*, and *Rag2^c/c* BCR-ABL1⁺ B-ALL mouse model.

**Figure supplement 2—source data 1.** Original file for the western blot analysis in *Figure 1—figure supplement 2A*.

**Figure supplement 3.** The genetic pathways in *Rag^f/f*, *Rag1^C/C*, and *Rag2^C/C* BCR-ABL1⁺ lymphocytes.

## The loss of non-core RAG regions highlights genomic DNA damage

The findings indicate that leukemic cells from three types of mice exhibited variable arrests at the large pre-B-cell stage, deviating from normal B-cell developmental trajectory. Typically, at this juncture, B cells initiate the degradation of Rag2 via the cyclin-dependent kinase cyclinA/Cdk2, leading to a downregulation of RAG activity. It is therefore crucial to explore the impact of deletions in non-core regions on the expression and functionality of RAG in these leukemic cells. Western blotting analysis showed that expression of Rag1^f/f, Rag1^c/c, Rag2^f/f, and Rag2^c/c proteins were present in GFP⁺CD19⁺ splenic leukemic cells with the indicated Rag1 and Rag2 genotypes (**Figure 3A**). Notably, we observed an upregulation expression of Rag1^c/c and Rag2^c/c in leukemic cells from *Rag1^c/c* or *Rag2^c/c* mice compared to those from *Rag^f/f* mice (**Figure 3A**, **Figure 3—figure supplement 1A**). The in vitro V(D)J recombination assay confirmed that different forms of RAG exhibited cleavage activity in *BCR-ABL1⁺* B-ALL (**Figure 3B**, **Figure 3—figure supplement 1B**). Moreover, we analyzed pediatric acute lymphoid leukemia patients from the TARGET cohort and found that high RAG expression was related to low survival in this disease (**Figure 3—figure supplement 2**).

To examine the potential correlation between aberrant RAG activity and increased DNA double-strand breaks (DSBs), we assessed levels of phosphorylated H2AX (γ-H2AX), a marker of DSB response, in leukemic cells from *Rag^f/f*, *Rag1^c/c*, and *Rag2^c/c* mice (gated on GFP⁺ cells). This evaluation aimed to gauge DNA DSBs and overall genomic instability. Flow cytometry analysis revealed elevated γ-H2AX levels in *Rag1^c/c* and *Rag2^c/c* leukemic cells compared to those from *Rag^f/f* (**Figure 3C**), indicating a more pronounced role of Rag^c/c in mediating somatic structural variants (SVs) in *BCR-ABL1⁺* B cells. These findings suggest enhanced expression of Rag1^c/c endonucleases in *Rag1^c/c* leukemic cells and increased DNA damage in cells lacking core RAG regions.

## Off-target recombination mediated by RAG in *BCR-ABL1⁺* B cells

Genome-wide sequencing and analysis were performed to compare somatic SVs in *BCR-ABL1⁺* B cells derived from *Rag^f/f*, *Rag1^c/c*, and *Rag2^c/c* mice. The leukemic cells were sequenced with an average coverage of 25× (**Supplementary file 1**). The SVs generated by RAG were screened based on two criteria: the presence of a CAC to the right (or GTG to the left) of both breakpoints, and its occurrence within 21 bp from the breakpoint (**Mijušković et al., 2015**). Further elaboration on these criteria can be found in **Figure 4—figure supplement 1**. Consequently, aberrant *V*-to-*V* junctions and *V* to intergenic regions were encompassed in five validated abnormal rearrangements at *Ig* loci in *Rag^c/c* leukemic mice (**Supplementary file 2**). Additionally, seven samples had 24 somatic SVs, with an average of 3.4 coding region mutations per sample (range of 0–9), which is consistent with the limited number of acquired somatic mutations observed in hematological cancers. The results of the study demonstrate that *Rag^f/f* cells had low SVs (0–1 per sample), *Rag1^c/c* cells exhibited higher SVs (6–9 per sample) while *Rag2^c/c* cells had moderate SVs incidence (1–4 per sample) (**Figure 4** and **Supplementary file 3**). These findings suggest that Rag^c/c may lead to an elevated off-target recombination, eventually posing a threat to the *BCR-ABL1⁺* B-cell genome.

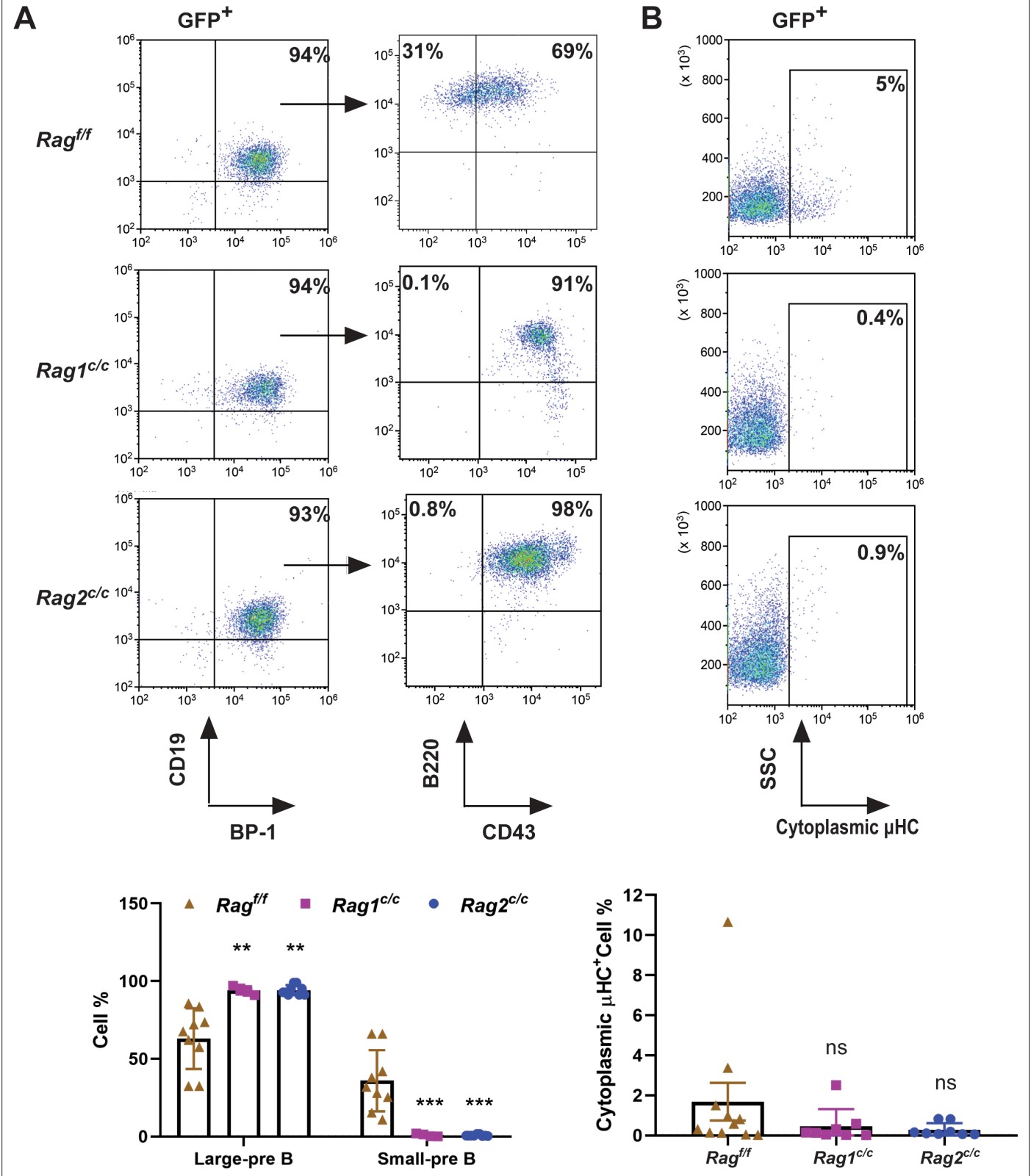

**Figure 2.** The non-core RAG region loss corresponds to a less mature cell surface phenotype. (**A**) Flow cytometry analysis of the B-cell markers CD19, BP-1, B220, and CD43 on *BCR-ABL1*-transformed *Rag^f/f^*, *Rag1^c/c^*, and *Rag2^c/c^* leukemic bone marrow cells. The percentages of each phase of the B-cell stage are summarized in the bottom graph (*Rag^f/f^*, n = 9, *Rag1^c/c^*, n = 4, *Rag2^c/c^*, n = 9; Large-preB, *Rag^f/f^*, vs *Rag1^c/c^*, p = 0.0349, *Rag^f/f^*, vs *Rag2^c/c^*, p = 0.0017; Small-pre-B, *Rag^f/f^*, vs *Rag1^c/c^*, p = 0.0141, *Rag^f/f^*, vs *Rag2^c/c^*, p = 0.0005). The expression of the cytoplasmic μ chain was analyzed by flow

*Figure 2 continued on next page*

*Figure 2 continued*

cytometry. Representative samples are shown in (**B**), and the results from multiple samples analyzed in independent experiments are summarized in the bottom graph as the fraction of cells expressing cytoplasmic factors (*Rag^{f/f}*, n = 11, *Rag1^{c/c}*, n = 9, *Rag2^{c/c}*, n = 8; *Rag^{f/f}* vs *Rag1^{c/c}*, p = 0.0312, *Rag^{f/f}*, vs *Rag2^{c/c}*, p = 0.0441). Error bars represent the mean ± standard deviation (s.d.), p values were calculated by Student's *t* test and **p < 0.01, ***p < 0.001.

The online version of this article includes the following source data and figure supplement(s) for figure 2:

**Figure supplement 1.** VDJ recombination in leukemic cells with different genetic backgrounds.

**Figure supplement 1—source data 1.** Original file for the PCR analysis in *Figure 2—figure supplement 1A*.

**Figure supplement 1—source data 2.** Original file for the PCR analysis in *Figure 2—figure supplement 1B*.

## Off-target V(D)J recombination characteristics in *BCR-ABL1^+* B cells

We further examined the characteristics of the identified SVs. Specifically, we analyzed the exon–intron distribution profiles of 41 breakpoints from 24 SVs through genome analysis. The results indicated that 57% of the breakpoints were located within the gene body, while 43% were enriched in the flanking sequences, the majority of which were identified as transcriptional regulatory sequences (*Figure 5A*). P and N nucleotides are recognized as distinctive characteristics of V(D)J recombination (*Repasky et al., 2004*). RSS-to-RSS and cRSS-to-cRSS recombination have P nucleotides lengths of 7 and 9, respectively, and N lengths of 5, so nucleotide lengths are basically the same during RSS-to-RSS and cRSS-to-cRSS recombination (*Figure 5B*). However, the frequency of P and N sequences in RSS-to-RSS recombination was 50%/50% (P/N), compared to 4%/8% (P/N) in cRSS-to-cRSS recombination (*Figure 5B*). This significant reduction in the frequency of P and N sequences suggests that DNA repair at off-target sites in *BCR-ABL1^+* B cells diverges from the classical V(D)J recombination repair process.

The hybrid joints were notably prevalent in *Rag1^{c/c}* and *Rag2^{c/c}* leukemic cells (93% and 100%, respectively), suggesting that the non-core regions of RAG may play a role in inhibiting harmful transposition events (*Figure 5C*). To evaluate the effect of deleting non-core RAG regions on the emergence of oncogenic mutations, we performed a comparative analysis of cancer-related genes across three types of leukemic cells. We found that *Rag1^{c/c}* leukemic cells harbored a significantly higher number of cancer genes compared to the other groups. This finding corresponds with the most aggressive leukemia phenotype observed in *Rag1^{c/c} BCR-ABL1^+* B-ALL mice and associated changes in their transcription profiles.

## The non-core regions have effects on RAG cleavage and off-target recombination size in *BCR-ABL1^+* B cells

Sequence logos were employed to visually contrast RSS and cRSSs within *Ig* and non-*Ig* loci, respectively. Notably, the RSS elements in *Ig* loci displayed a higher similarity to the canonical RSS, especially at critical functional sites. The initial four nucleotides (highlighted) of the canonical heptamer sequence <u>CACA</u>GTG were recognized as the cleavage site for Rag^{f/f}. Conversely, in leukemic cells, the cleavage site for Rag^{c/c} was pinpointed to the first three nucleotides, the CAC trinucleotide, of the heptamer sequence (*Figure 6A*). While both motifs (CAC and CACA) align with the highly conserved segment of the RSS heptamer sequence, differences in the cRSS sequences across off-target genes in both *Rag^{f/f}* and *Rag^{c/c}* mice suggest that deletion of RAG's non-core regions broadens the spectrum of off-target substrates in *BCR-ABL1^+* B cells.

Antigen receptor genes are assembled by large-scale deletions and inversions (*Teng et al., 2015*). The off-target recombination size was determined as the DNA fragment size spanning the two breakpoints. Our analysis demonstrated that both *Rag^{f/f}* and *Rag2^{c/c}* leukemic cells produced off-target recombinations with 100% and 92% of events, respectively, spanning over 10,000 bp in length. In contrast, *Rag1^{c/c}* leukemic cells showed only 6% of off-target recombinations exceeding 10,000 bp, with 48% under 1000 bp and 46% ranging between 1000 and 10,000 bp (*Figure 6B, C*). These findings suggest that the Rag1^{c/c} variant primarily facilitates smaller-scale off-target recombinations in *BCR-ABL1^+* B cells, highlighting the role of the non-core RAG1 region in influencing the extent of off-target recombination. The deletion of the non-core RAG1 region appears to constrict the size of off-target recombination, potentially contributing to the elevated frequency of off-target V(D)J recombination observed in *Rag1^{c/c}* leukemic cells (*Figure 6D*).

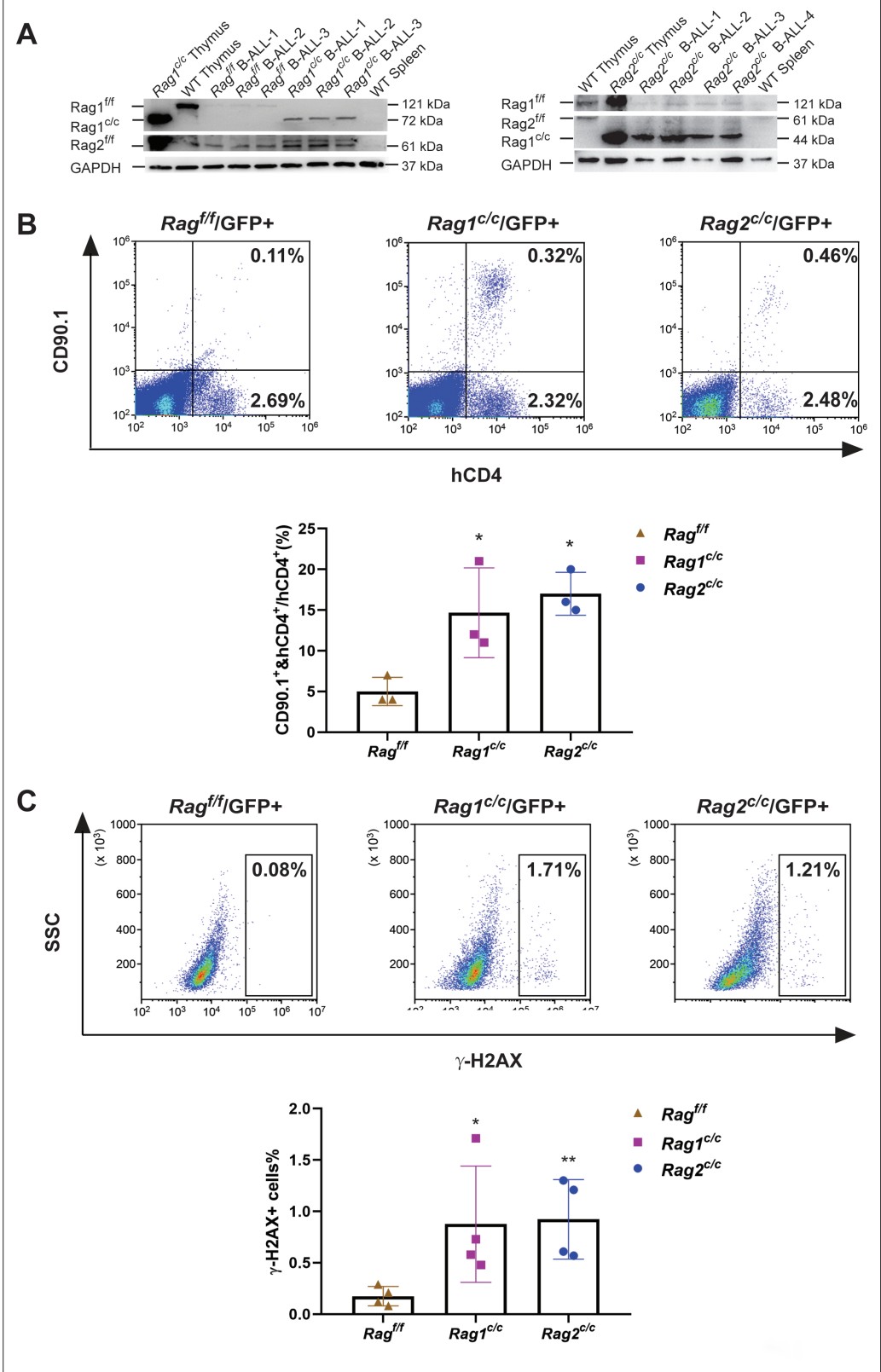

**Figure 3.** The non-core RAG region loss highlights genomic DNA damage. (**A**) Western blotting analysis showed RAG1 and RAG2 expression in GFP+CD19+ leukemic cells originating from *BCR-ABL1*+ B-ALL in different genetic backgrounds. The experiment was repeated under the same conditions three times. (**B**) Rearrangement substrate retrovirus was transduced into leukemic cells. Flow cytometry was used to analyze the percentage of CD90.1- and

*Figure 3 continued on next page*

*Figure 3 continued*

hCD4-positive cells, and the percentage populations are shown in the bottom graph (*Rag^{f/f}*, n = 3, *Rag1^{c/c}*, n = 3, *Rag2^{c/c}*, n = 3; *Rag^{f/f}*, vs *Rag1 ^{c/c}*, p = 0.0002, *Rag ^{f/f}*, vs *Rag2 ^{c/c}*, p = 0.5865). (**C**) Flow cytometry analysis of γ-H2AX levels in *Rag^{f/f}*, *Rag1^{c/c}*, and *Rag2^{c/c}* leukemic cells and the percentage of γ-H2AX-positive cell populations shown in the bottom graph (*Rag ^{f/f}*, n = 11, *Rag1^{c/c}*, n = 8, *Rag2^{c/c}*, n = 8; *Rag^{f/f}*, vs *Rag1^{c/c}*, p = 0.0505, *Rag^{f/f}*, vs *Rag2^{c/c}*, p = 0.0094). Error bars represent the mean ± standard deviation (s.d.), p values were calculated by Student's *t* test and \*p < 0.05, \*\*p < 0.01.

The online version of this article includes the following source data and figure supplement(s) for figure 3:

**Source data 1.** Original file for the western blot analysis in *Figure 3A*.

**Figure supplement 1.** RAG protein expression levels and schematic diagram of the recombinant substrate vector.

**Figure supplement 2.** The relationship of RAG1 mRNA levels and survival of pediatric acute lymphoid leukemia.

## Discussion

In this study, we have demonstrated that non-core region deletion of both Rag1 and Rag2 leads to accelerated development of leukemia and increased off-target V(D)J recombination in mouse models of *BCR-ABL1^+* B cells. Furthermore, we report reduced RAG^{c/c} cleavage accuracy and off-target recombination size in *Rag1^{c/c}* and *Rag2^{c/c}* leukemia cells, which might contribute to exacerbated off-target V(D)J recombination. These findings suggest that the non-core regions, particularly the non-core region of Rag1, play a crucial role in maintaining accuracy of V(D)J recombination and genomic stability in *BCR-ABL1^+* B cells.

Our findings suggest that leukemic cells with RAG^{c/c} regions exhibit increased production of hybrid joints, implying that non-core RAG regions might suppress the formation of these hybrid joints in vivo. Post-cleavage synaptic complexes (PSCs), comprising RAG proteins, coding ends, and RSS ends, are believed to have evolved to form with optimal conformation and/or stability for conventional coding and RSS end-joining (*Fugmann et al., 2000*; *Libri et al., 2021*). In contrast, RAG^{c/c} PSCs could promote RAG-mediated hybrid joints by facilitating closer proximity of coding and RSS ends or by increasing PSC stability. It is also conceivable that RAG recruit disassembly/remodeling factors to PSCs, a process that could allow NHEJ factors to complete the normal reaction (*Fugmann et al., 2000*). Conversely, RAG^{c/c} may have a diminished recruitment capacity due to changes in overall conformation or the absence of specific motifs, leading to more unstable PSCs and a heightened risk of accumulating incomplete hybrid joints (*Raghavan et al., 2006*; *Talukder et al., 2004*). Our data reveal that over 90% of junctions were hybrid joints in *Rag^{c/c}* leukemic cells, a frequency exceeding that reported in previous studies. This suggests that deficiencies in the NHEJ pathway could contribute to chromosomal instability and lymphomagenesis (*Gaymes et al., 2002*; *Rassool, 2003*; *Scully et al., 2019*; *Wiegmans et al., 2021*). Significantly, our analysis uncovered variations in the NHEJ repair pathway among leukemic cells from different genetic backgrounds, suggesting a potential aberrant expression of DNA repair pathways in *Rag^{c/c}* leukemic cells (*Figure 1—figure supplement 3B*). These findings highlight the potential of RAG^{c/c} to foster increased hybrid joint formation, especially when a normal pathway for efficient coding and RSS joining is compromised in an NHEJ-aberrant context.

Our data demonstrate that the deletion of the RAG1 non-core region results in more severe off-target V(D)J recombination compared to the deletion of the RAG2 non-core region. This observation is supported by the fact that the RAG1 terminus contains multiple zinc-binding motifs and ubiquitin ligase activity, which are known to enhance the efficiency of the rearrangement reaction (*Beilinson et al., 2021*; *Burn et al., 2022*). Furthermore, our research reveals that Rag1 expression persists in *BCR-ABL1^+* progenitor B cells, and deletion of the non-core region of Rag1 results in elevated expression of RAG in comparison to Rag^{f/f}. Consequently, as demonstrated in this study, Rag1^{c/c} from *BCR-ABL1^+* B leukemic mice is prone to generating off-target V(D)J recombination. The distinct function of Rag1's non-core region in thymic lymphomas of *Tp53^{−/−}* mice and *BCR-ABL1^+* B leukemic mice leads to dissimilar off-target activity of Rag1^{c/c} (*Mijušković et al., 2015*). Therefore, it would be intriguing to replicate these analyses across various subtypes of ALL to further investigate this phenomenon.

In human *ETV6-RUNX1* ALL, the *ETV6-RUNX1* fusion gene is believed to initiate prenatally, yet the disease remains clinically latent until critical secondary events occur, leading to leukemic transformation – 'pre-leukemia to leukemia' (*Mijušković et al., 2015*; *Bateman et al., 2010*). Genomic rearrangement, mediated by aberrant RAG recombinase activity, is a frequent driver of these secondary

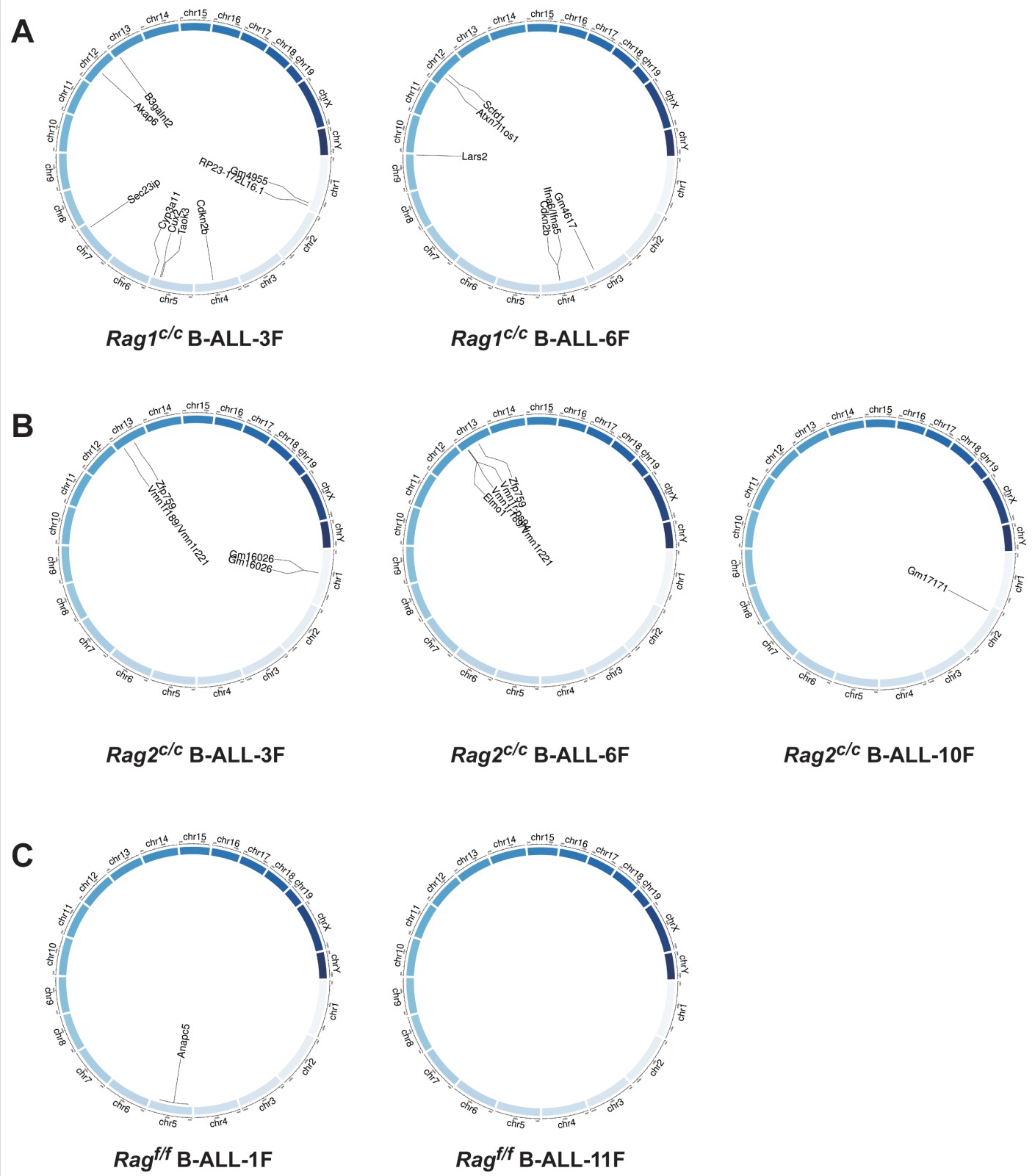

**Figure 4.** Structural alterations in *BCR-ABL1*[+] B lymphocytes. (**A–C**) Circos plot representation of all off-target recombination detected in the genome-wide analyses of *Rag*[f/f], *Rag1*[c/c], and *Rag2*[c/c] leukemic cells.

The online version of this article includes the following figure supplement(s) for figure 4:

**Figure supplement 1.** The criteria for identifying off-target recombination.

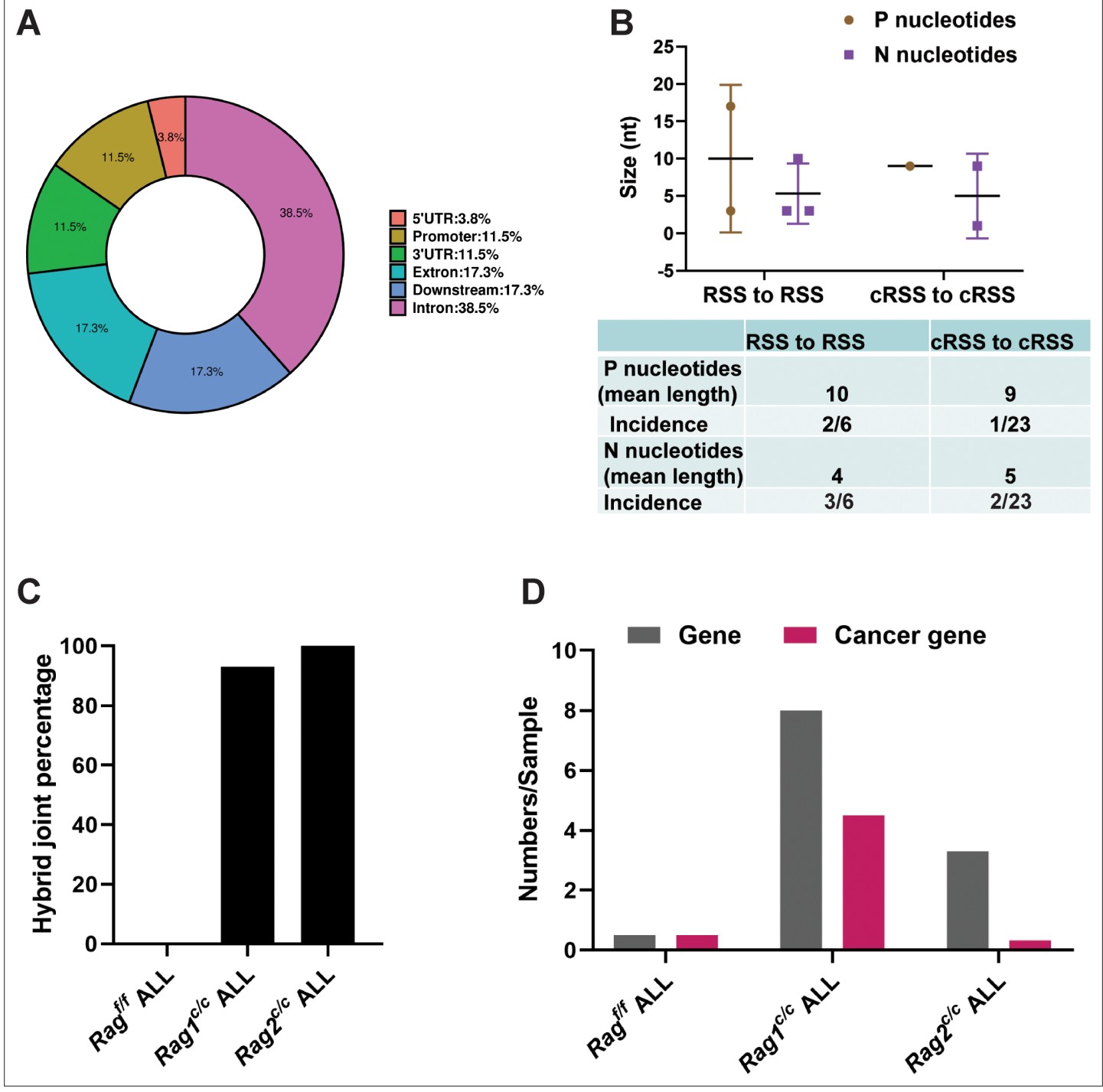

**Figure 5.** Overview and characteristics of off-target recombination in *BCR-ABL1*+ B-ALL leukemic cells from *Rag*^f/f and *Rag*^c/c mice. (**A**) Exon–intron distribution profiles of 41 breakpoints generated by 24 structural variations (SVs). Gene body includes exon (*n* = 9; 17.3%) and intron (*n* = 20; 38.5%). Flanking sequence includes 3'UTR (*n* = 6; 11.5%), 5'UTR (*n* = 2; 3.8%), promoter (*n* = 6; 11.5%), and downstream (*n* = 9; 17.3%). (**B**) The off-target recombination was filtered and verified by whole genomic sequence and PCR, respectively. P and N nucleotides of recombination signal sequence (RSS) to RSS and cryptic RSS (cRSS) to cRSS were calculated in *BCR-ABL1*+ B-ALL. (**C**) Hybrid joint percentage generated by either *Rag*^f/f, *Rag1*^c/c, or *Rag2*^c/c in *BCR-ABL1*+ B-ALL. It was 0, 100%, and 93% in *Rag*^f/f, *Rag1*^c/c, or *Rag2*^c/c leukemic cells, respectively. (**D**) The 24 off-target recombination genes were retrieved by COSMIC Cancer Gene Census (http://cancer.sanger.ac.uk/census/). 0.5 genes and 0.5 cancers gene average sample in *Rag*^f/f, leukemic cells; 8 genes and 4.5 cancer genes average sample in *Rag1*^c/c leukemic cells; 3.3 genes and 0.3 cancer genes average sample in *Rag2*^c/c leukemic cells.

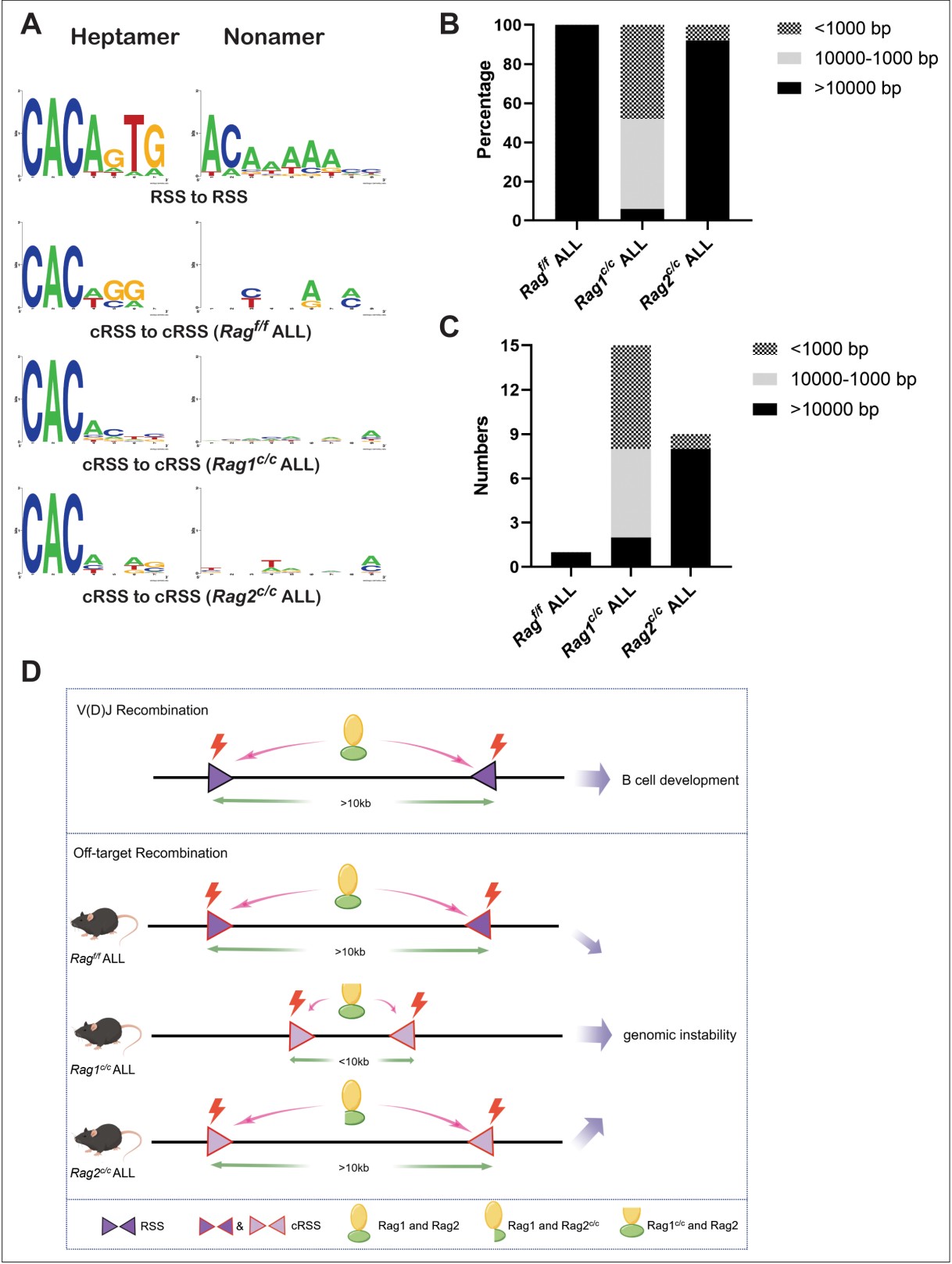

**Figure 6.** The non-core regions have effects on RAG-binding accuracy and recombinat size in *BCR-ABL1*+ B lymphocyte. (**A**) Sequence logos were used to compare the recombination signal sequence (RSS) and cryptic RSS (cRSS) in Ig loci and non-Ig loci. Top panel: V(D)J recombination at *Ig* locus; the next three panels: RAG-mediated off-target recombination at non-Ig locus from *Ragf/f*, *Rag1c/c*, and *Rag2c/c* leukemic cells, respectively. The scale of recombinant size was categorized into three ranges: <1000, 1000–10,000, and >10,000 bp. The distribution of different recombinant sizes in *Rag f/f*,

*Figure 6 continued on next page*

*Figure 6 continued*

*Rag1^{c/c}*, and *Rag2^{c/c}* leukemic cells was presented in (**B**), while the number of different recombinant sizes in *Rag ^{f/f}*, *Rag1^{c/c}*, and *Rag2^{c/c}* leukemic cells is displayed in (**C**). (**D**) A schematic depiction of the mechanism of cRAG-accelerated off-target V(D)J recombination was provided. Both RAG1 and RAG2's non-core region deletion decreases RAG-binding accuracy in *Rag1^{c/c}* and *Rag2^{c/c}*, *BCR-ABL1^+* B-ALL. Additionally, RAG1's non-core region deletion significantly reduces the size and scale of off-target V(D)J recombination in *Rag1^{c/c} BCR-ABL1^+* B-ALL.

events in *ETV6-RUNX1* ALL (*Chen et al., 2021*; *Wong et al., 2014*). In contrast, RAG-mediated off-target V(D)J recombination is also observed in *BCR-ABL1^+* B-ALL. These oncogenic SVs can also be considered as secondary events that promote the transition – 'leukemia to aggressive leukemia'. The enhancement of *BCR-ABL1^+* B-ALL deterioration and progression by RAG^{c/c} in mouse model was consistent with our previous study that RAG enhances BCR-ABL1-positive leukemic cell growth through its endonuclease activity (*Yuan et al., 2021*). Additionally, we showed that non-core RAG1 region deletion leads to increased Rag1^{c/c} expression and high RAG expression related to low survival in pediatric acute lymphoid leukemia. Therefore, more attention should be paid to the non-core RAG region mutation in *BCR-ABL1^+* B-ALL for the role of non-core region in leukemia suppression and off-target V(D)J recombination.

## Methods

### Mice

The C57BL/6 mice were purchased from the Experimental Animal Center of Xi'an Jiaotong University, while *Rag1^{c/c}* mice (amino acids 384–1040) and *Rag2^{c/c}* mice (amino acids 1–383) were obtained from Dr. David G. Schatz (Yale University, New Haven, CT, USA). The mice were bred and maintained in a specific pathogen-free environment at the Experimental Animal Center of Xi'an Jiaotong University. All animal-related procedures were in accordance with the guidelines approved by the Xi'an Jiaotong University Ethics Committee for Animal Experiments.

### Generation of retrovirus stocks

The pMSCV-BCR-BAL1-IRES-GFP vector is capable of co-expressing the human BCR-ABL1 fusion protein and GFP, while the pMSCV-GFP vector serves as a negative control by solely expressing GFP. To produce viral particles, 293T cells were transfected with either the MSCV-BCR-BAL1-IRES-GFP or MSCV-GFP vector, along with the packaging vector PKAT2, utilizing the X-tremeGENE HP DNA Transfection Reagent from Roche (Basel, Switzerland). After 48 hr, the viral supernatants were collected, filtered, and stored at −80°C.

### Bone marrow transduction and transplantation

Experiments were conducted using mice aged between 6 and 10 weeks. *BCR-ABL1^+* B-ALL murine model was induced by utilizing marrow from donor mice who had not undergone 5-fluorouracil treatment. The donor mice were euthanized through CO$_2$ asphyxiation, and the bone marrow was harvested by flushing the femur and tibia with a syringe and 26-gauge needle. Erythrocytes were not removed, and $1 \times 10^6$ cells per well were plated in 6-well plates. A single round of co-sedimentation with retroviral stock was performed in medium containing 5% WEHI-3B-conditioned medium and 10 ng/ml IL-7 (Peprotech, USA). After transduction, cells were either transplanted into syngeneic female recipient mice ($1 \times 10^6$ cells each) that had been lethally irradiated ($2 \times 450$ cGy), or cultured in RPMI-1640 (Hyclone, Logan, UT) medium supplemented with 10% fetal calf serum (Hyclone), 200 mmol/l L-glutamine, 50 mmol/l 2-mercaptoethanol (Sigma, St Louis, MO), and 1.0 mg/ml penicillin/streptomycin (Hyclone). Subsequently, recipient mice were monitored daily for indications of morbidity, weight loss, failure to thrive, and splenomegaly. Weekly assessment of peripheral blood GFP percentage was done using flow cytometry analysis of tail vein blood. Hematopoietic tissues and cells were utilized for histopathology, in vitro culture, FACS analysis, secondary transplantation, genomic DNA preparation, protein lysate preparation, or lineage analysis, contingent upon the unique characteristics of mice under study.

## Secondary transplants

Thawed bone marrow cells were sorted using a BD FACS Aria II (Becton Dickinson, San Jose, CA, USA). GFP-positive leukemic cells ($1 \times 10^6$, $1 \times 10^5$, $1 \times 10^4$, and $1 \times 10^3$) were then resuspended in 0.4 ml Hank's Balanced Salt Solution and intravenously administered to unirradiated syngeneic mice.

## Flow cytometry analysis and sorting

Bone marrow, spleen cells, and peripheral blood were harvested from leukemic mice. Red blood cells were eliminated using $NH_4Cl$ red blood cell lysis buffer, and the remaining nucleated cells were washed with cold phosphate-buffered saline (PBS). In order to conduct in vitro cell surface receptor staining, $1 \times 10^6$ cells were subjected to antibody staining for 20 min at 4°C in 1× PBS containing 3% BSA. Cells were then washed with 1× PBS and analyzed using a CytoFLEX Flow Cytometer (Beckman Coulter, Miami, FL) or sorted on a BD FACS Aria II. Apoptosis was analyzed by resuspending the cells in Binding Buffer (BD Biosciences, Baltimore, MD, USA), and subsequent labeling with anti-annexin V-AF647 antibody (BD Biosciences) and propidium iodide (BD Biosciences) for 15 min at room temperature. The lineage analysis was performed using the following antibodies, which were purchased from BD Biosciences: anti-BP-1-PerCP-Cy7, anti-CD19-PerCP-CyTM[5.5], anti-CD43-PE, anti-B220-APC, and anti-μHC-APC.

## BrdU incorporation and analysis

Cells obtained from primary leukemic mice were cultured in 6-well plates containing RPMI-1640 medium supplemented with 10% fetal bovine serum and 50 μg/ml BrdU. After incubation at 37°C for 30 min, the cells were harvested and intranuclearly stained with anti-BrdU and 7-AAD antibodies, following the manufacturer's instructions.

## The in vitro V(D)J recombination assay

The retroviral recombination substrate pINV-12/23 was introduced into primary leukemic cells utilizing X-treme GENE HP DNA Transfection Reagent (Roche). Recombination efficiency of pINV-12/23 was evaluated through flow cytometry analysis for mouse CD90 (mCD90) and hCD4 expression (*Yuan et al., 2021*).

## Western blotting analysis

Over $1 \times 10^6$ leukemic cells were centrifuged and washed with ice-cold PBS. The cells were then treated with ice-cold radioimmune precipitation assay buffer, consisting of 50 mM Tris–HCl (pH 7.4), 0.15 M NaCl, 1% Triton X-100, 0.5% NaDoc, 0.1% sodium dodecyl sulphide, 1 mM ethylene diamine tetraacetic acid, 1 mM phenylmethanesulphonyl fluoride (Amresco), and fresh protease inhibitor cocktail Pepstain A (Sigma). After sonication using a Bioruptor TMUCD-200 (Diagenode, Seraing, Belgium), the suspension was spined at $14,000 \times g$ for 3 min at 4°C. The total cell lysate was either utilized immediately or stored at −80°C. Protein concentrations were determined using DC Protein Assay (Bio-Rad Laboratories, Hercules, California, USA). Subsequently, the protein samples (20 μg) were incubated with α-RAG1 (mAb 23) and α-RAG2 (mAb 39) antibodies (*Teng et al., 2015*), with glyceraldehyde-3-phosphate dehydrogenase (GAPDH) serving as the loading control. The signal was further detected using secondary antibody of goat anti-rabbit IgG conjugated with horseradish peroxidase (Thermo Scientific, Waltham, MA). The band signal was developed with Immobilon Western Chemiluminescent HRP substrate (Millipore, Billerica, MA). The band development was analyzed using GEL-PRO ANALYZER software (Media Cybernetics, Bethesda, MD).

## Genomic PCR

Genomic PCR was performed in a 20-μl reaction containing 50 ng of genomic DNA, 0.2 μm of forward and reverse primer, and 10 μl Premix Ex Taq (TaKaRa, Shiga, Japan). Amplification conditions were as follows: 94°C for 5 min; 35 cycles of 30 s at 94°C, 30 s at 60°C and 1 min at 72°C; 72°C for 5 min (Bio-Rad, Hercules, CA). Genomic PCR was performed using the following primers:

DHL-5′-GGAATTCGMTTTTTGTSAAGGGATCTACTACTGTG-3′; JH3-5′-GTCTAGATTCTCAC-AAGAGTCCGATAGACCCTGG-3′; VHQ52-5′-CGGTACCAGACTGARCATCASCAAG

-GACAAYTCC-3′;    VH558-5′-CGAGCTCTCCARCACAGCCTWCATGCARCTCARC-3′;    VH7183-5′-
CGGTACCAAGAASAMCCTGTWCCTGCAAATGASC-3′ (*Schlissel et al., 1991*).

## RNA-seq library preparation and sequencing

GFP$^+$CD19$^+$ cells were sorted from the spleen of *Rag1$^{c/c}$* (n = 3, 1 × 10$^6$ cells/sample), *Rag2$^{c/c}$* (n = 3, 1 × 10$^6$ cells/sample), and *Rag$^{f/f}$* (n = 3, 1 × 10$^6$ cells/sample) *BCR-ABL1$^+$* B-ALL mice. Total RNA was extracted using Trizol reagent (Invitrogen, CA, USA) following the manufacturer's guidelines. RNA quantity and purity analysis were done using Bioanalyzer 2100 and RNA 6000 Nano LabChip Kit (Agilent, CA, USA) with RNA integrity number (RIN) >7.0. RNA-seq libraries were prepared by using 200 ng total RNA with TruSeq RNA sample prep kit (Illumina). Oligo(dT)-enriched mRNAs were fragmented randomly with fragmentation buffer, followed by first- and second-strand cDNA synthesis. After a series of terminal repair, the double-stranded cDNA library was obtained through PCR enrichment and size selection. cDNA libraries were sequenced with the Illumina Hiseq 2000 sequencer (Illumina HiSeq 2000 v4 Single-Read 50 bp) after pooling according to its expected data volume and effective concentration. Two biological replicates were performed in the RNA-seq analysis. Raw reads were then aligned to the mouse genome (GRCm38) using Tophat2 RNA-seq alignment software, and unique reads were retained to quantify gene expression counts from Tophat2 alignment files. The differentially expressed mRNAs and genes were selected with log2 (fold change)>1 or log2 (fold change) <−1 and with statistical significance (p value <0.05) by R package. Bioinformatic analysis was performed using the OmicStudio tools at https://www.omicstudio.cn/tool.

## Preparation of tumor DNA samples

GFP$^+$CD19$^+$ splenic cells, tail and kidney tissue were obtained from *Rag1$^{c/c}$*, *Rag2$^{c/c}$*, and *Rag$^{f/f}$* *BCR-ABL1$^+$* B-ALL mice, and genomic DNA was extracted using a TIANamp Genomic DNA Kit (TIAN-GEN-DP304). Subsequently, paired-end libraries were constructed from 1 µg of the initial genomic material using the TruSeq DNA v2 Sample Prep Kit (Illumina, #FC-121-2001) as per the manufacturer's instructions. The size distribution of the libraries was assessed using an Agilent 2100 Bioanalyzer (Agilent Technologies, #5067-4626), and the DNA concentration was quantified using a Qubit dsDNA HS Assay Kit (Life Technologies, #Q32851). The Illumina HiSeq 4000 was utilized to sequence the samples, with two to four lanes allocated for sequencing the tumor and one lane for the control DNA library of the kidney or liver, each with 150 bp paired end reads.

## Read alignment and SV calling

Fastq files were generated using Casava 1.8 (Illumina), and BWA 37 was employed to align the reads to mm9. PCR duplicates were eliminated using Picard's Mark Duplicates tool (source-forge.net/apps/mediawiki/picard). Our custom scripts (http://sourceforge.net/projects/svdetection) were utilized to eliminate BWA-designated concordant and read pairs with low BWA mapping quality scores. Intra- and inter-chromosomal rearrangements were identified using SV Detect from discordant, quality prefiltered read pairs. The mean insertion size and standard deviation for this analysis were obtained through Picard's InsertSizeMetrics tool (https://sourceforge.net/projects/picard/). Tumor-specific SVs were identified using the manta software (https://github.com/Illumina/manta; *Saunders, 2016*).

## Validation of high-confidence off-target candidates

The elimination of non-specific structural mutations from the kidney or tail was necessary for tumor-specific SVs identification. Subsequently, the method involving 21 bp CAC-to-breakpoint was employed to filter RAG-mediated off-target gene. The validation of high-confidence off-target candidates was carried out through PCR. Oligonucleotide primers were designed to hybridize within the 'linking' regions of SV Detect, in the appropriate orientation. The PCR product was subjected to Sanger sequencing and aligned to the mouse mm9 reference genome using BLAST (https://blast.ncbi.nlm.nih.gov/Blast.cgi).

## Statistics

Statistical analysis was conducted using SPSS 20.0 (IBM Corp) and GraphPad Prism 6.0 (GraphPad Software). Descriptive statistics were reported as means ± standard deviation for continuous variables. Statistical analyses were applied to biologically independent mice or technical replicates for

each experiment which was independently repeated at least three times. The equality of variances was assessed using Levene's test. Two-group comparisons, multiple group comparisons, and survival comparisons were performed using independent-samples $t$ test, one-way analyses of variance with post hoc Fisher's least significant difference (LSD) test, and log-rank Mantel–Cox analysis, respectively. Kaplan–Meier survival curves were utilized to depict the changes in survival rate over time. Statistical significance was set at $p < 0.05$.

## Acknowledgements

This study was supported by grants (no. 31170821, 31370874, and 81670157 to YJ) from the National Natural Science Foundation of China and by a grant (no. 2022JZ-44 to YJ) from the Natural Science Foundation of Shaanxi Province. The authors would like to thank Professor Shaoguang Li from the Division of Hematology/Oncology, University of Massachusetts Medical School, for providing the MSCV-BCR-BAL1-IRES-GFP construct. The authors would also like to thank Mr. Xiaofei Wang (Xi'an Jiaotong University Health Science Centre) for providing expert technical assistance with cell sorting.

---

## Additional information

### Funding

| Funder | Grant reference number | Author |
|---|---|---|
| National Natural Science Foundation of China | 31170821 | Yanhong Ji |
| Natural Science Foundation of Shaanxi Province | 2022JZ-44 | Yanhong Ji |
| National Natural Science Foundation of China | 31370874 | Yanhong Ji |
| National Natural Science Foundation of China | 81670157 | Yanhong Ji |
| Xi 'an Jiaotong University Medical "Basic - Clinical" Fusion Innovation Project | YXJLRH2022012 | Yanhong Ji |
| Xi 'an Jiaotong University Medical "Basic - Clinical" Fusion Innovation Project | YJSCX-2024-007 | Yanhong Ji |

The funders had no role in study design, data collection, and interpretation, or the decision to submit the work for publication.

### Author contributions

Xiaozhuo Yu, Conceptualization, Validation, Visualization, Methodology, Writing – original draft; Wen Zhou, Validation, Visualization, Methodology; Xiaodong Chen, Validation, Methodology, Writing – review and editing; Shunyu He, Data curation; Mengting Qin, Data curation, Methodology; Meng Yuan, Validation, Writing – review and editing; Yang Wang, Software, Writing – review and editing; Woodvine Otieno Odhiambo, Writing – review and editing; Yinsha Miao, Writing – original draft; Yanhong Ji, Conceptualization, Formal analysis, Investigation, Methodology, Writing – original draft, Writing – review and editing

### Author ORCIDs

Xiaozhuo Yu ⓘ http://orcid.org/0009-0006-7565-8941
Yanhong Ji ⓘ http://orcid.org/0000-0003-4144-4786

### Ethics

All animal experiments were performed with the approval of the Institutional Animal Care and Use Committee of Xi'an Jiaotong University (under reference XJTU#20202272).

Reviewer #1 (Public Review): https://doi.org/10.7554/eLife.91030.3.sa1
Author response https://doi.org/10.7554/eLife.91030.3.sa2

## Additional files

### Supplementary files
- Supplementary file 1. Sequencing statistics.
- Supplementary file 2. Abnormal rearrangements at Ig.
- Supplementary file 3. RAG-mediated off-target genes.
- MDAR checklist

### Data availability
Files containing sequencing reads were deposited in the National Institutes of Health Sequence Read Archive under PRJNA1116598.

The following dataset was generated:

| Author(s) | Year | Dataset title | Dataset URL | Database and Identifier |
|---|---|---|---|---|
| Ji Y | 2024 | RAG1 and RAG2 Non-core Regions Are Implicated in Leu-kemogenesis and Off-target V(D)J Recombination in BCR-ABL1-driven B-cell Lineage Lymphoblastic Leukemia | https://www.ncbi.nlm.nih.gov/bioproject/PRJNA1116598/ | NCBI BioProject, PRJNA1116598 |

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
