## [Editor Report · eLife assessment]

Using a set of animal models, this **valuable** paper shows tumor suppressive function of the non-core regions of RAG1/2 recombinases. The conclusions are supported by **solid** evidence.

---

## [Referee Report · Reviewer #1 (Public Review)]

Summary:

In this report, Yu et al ascribe potential tumor suppressive functions to the non-core regions of RAG1/2 recombinases. Using a well-established BCR-ABL oncogene-driven system, the authors model the development of B cell acute lymphoblastic leukemia in mice and found that RAG mutants lacking non-core regions show accelerated leukemogenesis. They further report that the loss of non-core regions of RAG1/2 increases genomic instability, possibly caused by increased off-target recombination of aberrant RAG-induced breaks. The authors conclude that the non-core regions of RAG1 in particular not only increases the fidelity of VDJ recombination, but may also influence the recombination "range" of off-target joints, and that in the absence of the non-core regions, mutant RAG1/2 (termed cRAGs) catalyze high levels of off-target recombination leading to the development of aggressive leukemia.

Strengths:

The authors used a genetically defined oncogene-driven model to study the effect of RAG non-core regions have on leukemogenesis. The animal studies were well performed and generally included a good number of mice. Therefore, the finding that cRAG expression led to development of more aggressive BCR-ABL+ leukemia compared to fRAG is solid. The authors also present some nice analyses that characterize the (genomic) nature of aggressive leukemia that develop in the absence of RAG non-core regions.

Weaknesses:

The paper relies on cRAG1/2 overexpression, an experimental limitation that needs to be taken into consideration when extrapolating the physiological relevance of the findings.

---

## [Author Response]

The following is the authors’ response to the original reviews.

We would like to thank all of the reviewers for their helpful and the effort they made in reading and evaluating our manuscript. In response to them, we have made major changes to the text and figures and performed substantial new experiments. These new data and changes to the text and figures have substantially strengthened the manuscript. We believe that the manuscript is now very strong in both its impact and scope and we hope that reviewers will find it suitable for publication in eLife

A point-by-point response to the reviewers' specific comments is provided below.

**Public Reviews:**

**Reviewer #1 (Public Review):**
Summary:In this report, Yu et al ascribe potential tumor suppressive functions to the non-core regions of RAG1/2 recombinases. Using a well-established BCR-ABL oncogene-driven system, the authors model the development of B cell acute lymphoblastic leukemia in mice and found that RAG mutants lacking non-core regions show accelerated leukemogenesis. They further report that the loss of non-core regions of RAG1/2 increases genomic instability, possibly caused by increased off-target recombination of aberrant RAG-induced breaks. The authors conclude that the non-core regions of RAG1 in particular not only increase the fidelity of VDJ recombination, but may also influence the recombination "range" of off-target joints, and that in the absence of the non-core regions, mutant RAG1/2 (termed cRAGs) catalyze high levels of off-target recombination leading to the development of aggressive leukemia.Strengths:The authors used a genetically defined oncogene-driven model to study the effect of RAG non-core regions on leukemogenesis. The animal studies were well performed and generally included a good number of mice. Therefore, the finding that cRAG expression led to the development of more aggressive BCR-ABL+ leukemia compared to fRAG is solid.Weaknesses:In general, I find the mechanistic explanation offered by the authors to explain how the non-core regions of RAG1/2 suppress leukemogenesis to be less convincing. My main concern is that cRAG1 and cRAG2 are overexpressed relative to fRAG1/2. This raises the possibility that the observed increased aggressiveness of cRAG tumors compared to fRAG tumors could be solely due to cRAG1/2 overexpression, rather than any intrinsic differences in the activity of cRAG1/2 vs fRAG1/2; and indeed, the authors allude to this possibility in Fig S8, where it was shown that elevated expression of RAG (i.e. fRAG) correlated with decreased survival in pediatric ALL. Although it doesn't mean the authors' assertions are incorrect, this potential caveat should nevertheless be discussed.

We appreciate the valuable suggestions from the reviewer. BCR-ABL1+ B-ALL is characterized by halted early B-lineage differentiation. In BCR-ABL1+ B cells, RAG recombinases are highly expressed, leading to the inactivation of genes that encode essential transcription factors for B-lineage differentiation. This results in cells being trapped within the precursor compartment, thereby elevating RAG gene expression. Our interpretation of the data suggests that, in BCR-ABL1+ B-ALL mouse models, the high expression of both cRAG and fRAG and the deletion of the non-core regions influence the precision of RAG targeting within the genome. This causes more genomic damage in cRAG tumors than in fRAG tumors, consequently leading to the observed increased aggressiveness of cRAG tumors compared to fRAG tumors. We discussed the issues on Page 12, lines 295-307 in the revised manuscript.

Some of the conclusions drawn were not supported by the data.(1) I'm not sure that the authors can conclude based on μHC expression that there is a loss of pre-BCR checkpoint in cRAG tumors. In fact, Fig. 2B showed that the differences are not statistically significant overall, and more importantly, μHC expression should be detectable in small pre-B cells (CD43-). This is also corroborated by the authors' analysis of VDJ rearrangements, showing that it has occurred at the H chain locus in cRAG cells.

We appreciate the insightful comment from the reviewer. Upon reevaluation of the data presented in Fig. 2B, we identified and rectified certain errors. The revised analysis now shows that the differences in μHC expression are statistically significant. This significant expression of μHC in fRAG leukemic cells implies that these cells may progress further in differentiation, potentially acquiring an immune phenotype. These modifications have been incorporated into the manuscript on page 7, lines 153-156 in the revised manuscript.

(2) The authors found a high degree of polyclonal VDJ rearrangements in fRAG tumor cells but a much more limited oligoclonal VDJ repertoire in cRAG tumors. They concluded that this explains why cRAG tumors are more aggressive because BCR-ABL induced leukemia requires secondary oncogenic hits, resulting in the outgrowth of a few dominant clones (Page 19, lines 381-398). I'm not sure this is necessarily a causal relationship since we don't know if the oligoclonality of cRAG tumors is due to selection based on oncogenic potential or if it may actually reflect a more restricted usage of different VDJ gene segments during rearrangement.

Thank you for your insightful comments and questions regarding the relationship between the oligoclonality of V(D)J rearrangements and the aggressiveness of cRAG tumors. You raise an important point regarding whether the observed oligoclonality is a result of selective pressure favoring clones with specific oncogenic potential, or if it reflects inherent limitations in V(D)J segment usage during rearrangement in cRAG models. In our study, we observed a marked difference in the V(D)J rearrangement patterns between fRAG and cRAG tumor cells, with cRAG tumors exhibiting a more limited, oligoclonal repertoire. This observation led us to speculate that the aggressive nature of cRAG tumors might be linked to a selective advantage conferred by specific V(D)J rearrangements that cooperate with the BCR-ABL1 oncogene to drive leukemogenesis. However, we acknowledge that our current data do not definitively establish a causal relationship between oligoclonality and tumor aggressiveness. The restricted V(D)J repertoire in cRAG tumors could indeed be due to a more constrained rearrangement process, possibly influenced by the altered expression or function of RAG1/2 in the absence of non-core regions. This could limit the diversity of V(D)J rearrangements, leading to the emergence of a few dominant clones not necessarily because they have greater oncogenic potential, but because of a narrowed field of rearrangement possibilities.

To address this question more thoroughly, future studies could examine the functional consequences of specific V(D)J rearrangements found in dominant cRAG tumor clones. This could include assessing the oncogenic potential of these rearrangements in isolation and in cooperation with BCR-ABL1, as well as exploring the mechanistic basis for the restricted V(D)J repertoire. Such studies would provide deeper insight into the interplay between RAG-mediated recombination, clonal selection, and leukemogenesis in BCR-ABL1+ B-ALL.

We appreciate your feedback on this matter and agree that further investigation is required to unravel the precise relationship between V(D)J rearrangement diversity and leukemic progression in cRAG models. We have revised our discussion to reflect these considerations and to clarify the speculative nature of our conclusions regarding the link between oligoclonality and tumor aggressiveness. We added more discussion on this issue on Page 7, lines 166-170 in the revised manuscript.

(3) What constitutes a cancer gene can be highly context- and tissue-dependent. Given that there is no additional information on how any putative cancer gene was disrupted (e.g., truncation of regulatory or coding regions), it is not possible to infer whether increased off-target cRAG activity really directly contributed to the increased aggressiveness of leukemia.

We totally agree you raised the issues. In Supplementary Table 3, we have presented data on off-target gene disruptions, specifically in introns, exons, downstream regions, promoters, 3' UTRs, and 5' UTRs. However, this dataset alone does not suffice to conclusively determine whether the increased off-target activity of cRAG directly influences the heightened aggressiveness of leukemia. To bridge this knowledge gap, our future research will extend to include both knockout and overexpression experiments targeting these off-target genes.

(4) Fig. 6A, it seems that it is really the first four nucleotide (CACA) that determines fRAG binding and the first three (CAC) that determine cRAG binding, as opposed to five for fRAG and four for cRAG, as the author wrote (page 24, lines 493-497).

We thank the reviewer for the insightful comment. In response, we have revised the text to accurately reflect the nucleotide sequences responsible for RAG binding and cleavage. Specifically, we now clarify that the first four nucleotides (CACA) are crucial for fRAG binding and cleavage, while the initial three nucleotides (CAC) are essential for cRAG binding and cleavage. These updates have been made on page 10, lines 242-245 of the revised manuscript.

(5) Fig S3B, I don't really see why "significant variations in NHEJ" would necessarily equate "aberrant expression of DNA repair pathways in cRAG leukemic cells". This is purely speculative. Since it has been reported previously that alt-EJ/MMEJ can join off target RAG breaks, do the authors detect high levels of microhomology usage at break points in cRAG tumors?

We appreciate the reviewer's comment. Currently, we have not observed microhomology usage at breakpoints in cRAG tumors. We plan to address this aspect in a future, more detailed study. Regarding the 'aberrant expression of DNA repair pathways in cRAG leukemic cells, we acknowledge that this is speculative. Therefore, we have carefully rephrased this to 'suggesting a potential aberrant expression of DNA repair pathways in cRAG leukemic cells.' This modification is reflected on page 12, lines 290-291 of the revised manuscript.

(6) Fig. S7, CDKN2B inhibits CDK4/6 activation by cyclin D, but I don't think it has been shown to regulate CDK6 mRNA expression. The increase in CDK6 mRNA likely just reflects a more proliferative tumor but may have nothing to do with CDKN2B deletion in cRAG1 tumors.

We fully concur with the reviewer's comment. We have deleted this inappropriate part from the text.

Insufficient details in some figures. For instance, Fig. 1A, please include statistics in the plot showing a comparison of fRAG vs cRAG1, fRAG vs cRAG2, cRAG1 vs cRAG2. As of now, there's a single p-value (0.0425) stated in the main text and the legend but why is there only one p-value when fRAG is compared to cRAG1 or cRAG2? Similarly, the authors wrote "median survival days 11-26, 10-16, 11-21 days, P < 0.0023-0.0299, Fig. S2B." However, it is difficult for me to figure out what are the numbers referring to. For instance, is 11-26 referring to median survival of fRAG inoculated with three different concentrations of GFP+ leukemic cells or is 11-26 referring to median survival of fRAG, cRAG1, cRAG2 inoculated with 10^5 cells? It would be much clearer if the authors can provide the numbers for each pair-wise comparison, if not in the main text, then at least in the figure legend. In Fig. 5A-B, do the plots depict SVs in cRAG tumors or both cRAG and fRAG cells? Also in Fig. 5, why did 24 SVs give rise to 42 breakpoints, and not 48? Doesn't it take 2 breaks to accomplish rearrangement? In Fig. 6B-C, it is not clear how the recombination sizes were calculated. In the examples shown in Fig. 4, only cRAG1 tumors show intra-chromosomal joins (chr 12), while fRAG and cRAG2 tumors show exclusively inter-chromosomal joins.

We appreciate the reviewer's feedback and have made the following revisions:

(1) The text has been adjusted to rectify the previously mentioned error in the figure legends (page 1, lines 5-6).

(2) We have clarified the intended message in the revised text (page 6, lines 129-130) and the figure legend (page 4-5, lines 107-113) for greater precision.

(3) Figure 5A-B now presents an overview of all structural variants (SVs) identified in both cRAG and fRAG cells, offering a comprehensive comparison.

(4) Among the analyzed SVs, 24 generated a total of 48 breakpoints, with 41 occurring within gene bodies and the remaining 7 in adjacent flanking sequences. This informs our exon-intron distribution profile analysis.

(5) We have defined recombination sizes as ‘the DNA fragment size spanning the two breakpoints’ for clarity (page 10, lines 251-252).

(6) All off-target recombinations identified in the genome-wide analyses of fRAG, cRAG1, and cRAG2 leukemic cells were determined to be intra-chromosomal joins, highlighting their specific nature within the genomic context.

Insufficient details on certain reagents/methods. For instance, are the cRAG1/2 mice of the same genetic background as fRAG mice (C57BL/6 WT)? On Page 23, line 481, what is a cancer gene? How are they defined? In Fig. 3C, are the FACS plots gated on intact cells? Since apoptotic cells show high levels of gH2AX, I'm surprised that the fraction of gH2AX+ cells is so much lower in fRAG tumors compared to cRAG tumors. The in vitro VDJ assay shown in Fig 3B is not described in the Method section (although it is described in Fig S5b). Fig. 5A-B, do the plots depict SVs in cRAG tumors or both cRAG and fRAG cells?

We are grateful for the reviewer's feedback and have incorporated their insights as follows:

(1) We clarify that both cRAG1/2 and fRAG mice share the same genetic background, specifically the C57BL/6 WT strain, ensuring consistency across experimental models.

(2) We define a 'cancer gene' as one harboring somatic mutations implicated in cancer. To support our analysis, we refer to the Catalogue Of Somatic Mutations In Cancer (COSMIC) at http://cancer.sanger.ac.uk/cosmic. COSMIC serves as the most extensive repository for understanding the role of somatic mutations in human cancers.

(3) Upon thorough review of the raw data for γ-H2AX and the fluorescence-activated cell sorting (FACS) plots gated on intact cells, we propose that the observed discrepancies might stem from the limited sensitivity of the γ-H2AX flow cytometry detection method. This insight prompts our commitment to employing more efficient detection methodologies in forthcoming studies.

(4) Detailed procedures for the in vitro V(D)J recombination assay have been included in the Methods section (page 15, lines 384-388) to enhance the manuscript's comprehensiveness and reproducibility.

(5) The presented plots offer a comprehensive overview of structural variants (SVs) identified in both cRAG and fRAG cells, providing a holistic view of the genomic landscape across different models.

**Reviewer #3 (Public Review):**
Summary:In the manuscript, the authors summarized and introduced the correlation between the non-core regions of RAG1 and RAG2 in BCR-ABL1+acute B lymphoblastic leukemia and off-target recombination which has certain innovative and clinical significance.
**Recommendations For The Authors:**

**Reviewer #1 (Recommendations For The Authors):**
I would suggest that the authors tone down some of their conclusions, which are not necessarily supported by their own data. in addition, there are some minor mistakes in figure assembly/presentation. For instance, I believe that the axes labels in Fig. 1E were flipped. BrdU should be on y-axis and 7-AAD on the x-axis. Fig. 3B, the y-axis contains a typo, it should be "CD90.1..." and not "D90.1...". In Fig. 5C, the numbers seem to be flipped, with 93% corresponding to cRAG1 and 100% to cRAG2 (compare with the description on page 23, lines 474-475). Fig. 5C, y-axis, "hybrid" is a typo. Page 3, line 59: The abbreviation of RSS has already been described earlier (p4, line 53).

We thank the reviewer for these suggestions. We carefully checked the raw data and corrected these mistakes in the revised manuscript.

Page 3, line 63: "signal" segment (commonly referred to as signal ends), not "signaling" segment.

We have changed “signaling segment” to “signal ends in the revised manuscript. (page 3, lines 54-55)

Page 3, lines 64-65: VDJ recombination promotes the development of both B and T cells, and aberrant recombination can cause both B and T cell lymphomas.

The statement about the role of V(D)J recombination in B and T cell development and its link to lymphomagenesis is grounded in a substantial body of research. Theoretical frameworks and empirical studies delineate how aberrations in the recombination process can lead to genomic instability, potentially triggering oncogenic events. This connection is extensively documented in immunology and oncology literature, illustrating the critical balance between necessary genetic rearrangements for immune diversity and the risk of malignancy when these processes are dysregulated (Thomson, et al.,2020; Mendes, et al.,2014; Onozawa and Aplan,2012).

Page 4, line 72: "recombinant dispensability" is not a commonly used phrase. Do the authors mean the say that the non-core regions of RAG1/2 are not strictly required for VDJ recombination?

We thank the reviewers for their insightful suggestion. We have revised the sentence to read, 'Although the non-core regions of RAG1/2 are not essential for V(D)J recombination, the evolutionary conservation of these regions suggests their potential significance in vivo, possibly affecting RAG activity and expression in both quantitative and qualitative manners.' This revision appears on page 3, lines 61-62, in the revised manuscript.

Fig. 4. It would have been nice to show at least one more cRAG1 tumor circus plot.

We appreciate the reviewer's comment and concur with the suggestion. In future sequencing experiments, we will consider including additional replicates. However, due to time and financial constraints, the current sequencing effort was limited to a maximum of three replicates.

**Reviewer #3 (Recommendations For The Authors):**
In the manuscript, the authors summarized and introduced the correlation between the non-core regions of RAG1 and RAG2 in BCR-ABL1+acute B lymphoblastic leukemia and off-target recombination which has certain innovative and clinical significance. The following issues need to be addressed by the authors.(1) Authors should check and review extensively for improvements to the use of English.

We thank the reviewer for their comment. With assistance from a native English speaker, we have carefully revised the manuscript to enhance its readability.

(2) Authors should revise the conclusion so that the above can be clearly reviewed and summarized.

The conclusion has been partially revised in the revised manuscript.

(3) The article should state that the experiment was independently repeated three times.

The experiment was repeated under the same conditions three times and the information has been descripted in Statistics section on page 19, lines 473-475 in the revised manuscript.

(4) The article will be more convincing if it uses references in the last 5 years.

We are grateful to the reviewer for their guidance in enhancing our manuscript. We have incorporated additional references from the past five years in the revised version.

(5) Additional experiments are suggested to elucidate the molecular mechanisms related to off-target recombination.

We thank the reviewer for this suggestion. In future experiments, we plan to perform ChIP-seq analysis to investigate the relationship between chromatin accessibility and off-target effects, as well as to examine the impact of knocking out and overexpressing off-target genes on cancer development and progression.

(6) It is suggested to further analyze the effect of the absence of non-core RAG region on the differentiation and development of peripheral B cells in mice by flow analysis and expression of B1 and B2.

Thank you very much for highlighting this crucial issue. FACS analysis was performed, revealing that leukemia cells in peripheral B cells in mice did not express CD5. The data are presented as follows:

**Author response image 1. sa2fig1:** 

(7) Fig3A should have three biological replicates and the molecular weight should be labeled on the right side of the strip.

Thank you for this suggestion. The experiment was independently repeated three times, and the molecular weights have been labeled on the right side of the bands in the revised version

References:

Mendes RD, Sarmento LM, Canté-Barrett K, Zuurbier L, Buijs-Gladdines JG, Póvoa V, Smits WK, Abecasis M, Yunes JA, Sonneveld E, Horstmann MA, Pieters R, Barata JT, Meijerink JP. 2014. PTEN microdeletions in T-cell acute lymphoblastic leukemia are caused by illegitimate RAG-mediated recombination events. BLOOD 124:567-578. doi:10.1182/blood-2014-03-562751

Onozawa M, Aplan PD. 2012. Illegitimate V(D)J recombination involving nonantigen receptor loci in lymphoid malignancy. Genes Chromosomes Cancer 51:525-535. doi:10.1002/gcc.21942

Thomson DW, Shahrin NH, Wang P, Wadham C, Shanmuganathan N, Scott HS, Dinger ME, Hughes TP, Schreiber AW, Branford S. 2020. Aberrant RAG-mediated recombination contributes to multiple structural rearrangements in lymphoid blast crisis of chronic myeloid leukemia. LEUKEMIA 34:2051-2063. doi:10.1038/s41375-020-0751-y